



# The Wave Geometry of Final Stratospheric Warming Events

Amy H. Butler[1] and Daniela I.V. Domeisen[2]

[1]National Oceanic and Atmospheric Administration, Chemical Sciences Laboratory, Boulder, CO, USA
[2]Institute for Atmospheric and Climate Science, ETH Zürich, Switzerland

**Correspondence:** Amy H. Butler (amy.butler@noaa.gov)

**Abstract.** Every spring, the stratospheric polar vortex transitions from its westerly wintertime state to its easterly summertime state due to seasonal changes in incoming solar radiation, an event known as the "final stratospheric warming" (FSW). While FSWs tend to be less abrupt than reversals of the boreal polar vortex in midwinter, known as sudden stratospheric warming (SSW) events, their timing and characteristics can be significantly modulated by atmospheric planetary-scale waves. Just like

SSWs, FSWs have been found to have predictable surface impacts. While SSWs are commonly classified according to their wave geometry, either by how the vortex evolves (whether the vortex displaces off the pole or splits into two vortices) or by the dominant wavenumber of the vortex just prior to the SSW (wave-1 versus wave-2), little is known about the wave geometry of FSW events. We here show that FSW events for both hemispheres in most cases exhibit a clear wave geometry. Most FSWs can be classified into wave-1 or wave-2 events, but wave-3 also plays a significant role in both hemispheres. Additionally, we find

that in the Northern Hemisphere, wave-2 events are more likely to occur later in the spring, while in the Southern Hemisphere, wave-1 or wave-2 events show no clear preference in timing. The FSW enhances total column ozone over the pole of both hemispheres during spring, but the spatial distribution of ozone anomalies can be influenced by the wave geometry and the timing of the event. We also describe the stratosphere's downward influence on surface weather following wave-1 and wave-2 FSW events. Significant differences between the tropospheric response to wave-1 and wave-2 FSW events occur over North

America and over the Southern Ocean, while no significant differences are found over the North Atlantic region, Europe, and Antarctica.

## 1  Introduction

The polar stratosphere exhibits a distinct seasonal cycle featuring a wintertime polar vortex, that is, strong circumpolar westerly

winds that form in late summer and decay the following spring, which is ultimately due to the seasonal cycle of incoming solar radiation. While the formation of the polar vortex occurs very predictably each year in late summer of both hemispheres (late August in the Northern and mid-February in the Southern Hemisphere), the timing of the spring weakening of the vortex, the so-called final stratospheric warming (FSW) event, is more variable (Black et al., 2006; Black and McDaniel, 2007a). The FSW marks the reversal of the climatological winter westerlies to summer easterlies in the stratosphere, and its timing varies





by up to two months in the Northern Hemisphere (NH) and by more than one month in the Southern Hemisphere (SH) due to upward-propagating wave disturbances from the troposphere that can disrupt the vortex ahead of its radiatively-driven decay (Waugh et al., 1999; Black and McDaniel, 2007a, b). FSWs thus share many characteristics with dynamically-driven midwinter disruptions of the polar vortex, spectacular events called sudden stratospheric warmings (SSW), in which the polar stratosphere rapidly warms and the polar vortex winds reverse. However, FSW events are driven by a combination of wave-induced and

radiative processes (Salby and Callaghan, 2007), and thus occur every spring in both hemispheres, while the occurrence of major SSW events is largely limited to the NH, with a notable exception in the SH spring of 2002 (e.g. Charlton et al., 2005). In the NH, SSWs on average occur about six times per decade (Charlton and Polvani, 2007) with strong decadal variability (Reichler et al., 2012; Domeisen, 2019). Further notable differences between the NH and the SH include a longer lifespan of the SH vortex and a stronger distortion and displacement from the pole of the NH vortex (Waugh and Randel, 1999).

In the SH spring, the timing of the FSW is modulated by feedbacks between chemical stratospheric ozone loss and the circulation (Solomon et al., 2014). The SH spring vortex is climatologically stronger and more stable compared to the NH, allowing annual conditions ideal for rapid destruction of ozone by atmospheric chlorofluorocarbons, known as the ozone hole (Solomon, 1999). As sunlight returns to the South pole every year in late September, a cascade of chemical reactions rapidly destroys stratospheric ozone, which further cools and strengthens the polar vortex and allows the vortex to persist longer.

Because years with larger ozone loss tend to have later FSWs in the SH (Zhang et al., 2017), the SH exhibits a long-term trend in the timing of FSW events that is linked to ozone depletion (e.g. Zhou et al., 2000; Haigh and Roscoe, 2009; Sheshadri et al., 2014). In the NH, where spring temperatures are rarely cold enough to support chemical reactions for rapid ozone loss, the persistence of the vortex in the NH spring is more closely linked to interannual variations in tropospheric wave forcing than to feedbacks with stratospheric ozone (Chipperfield and Jones, 1999; Newman et al., 2001; Savenkova et al., 2012). Nevertheless

certain boreal springs, as in 1997 and 2020, have been characterized by a persistent polar vortex associated with extreme Arctic ozone loss (Coy et al., 1997; Lawrence et al., 2020). The timing of the FSW in both hemispheres can have significant influence on the transport and mixing of stratospheric ozone (Rood and Schoeberl, 1983; Manney and Lawrence, 2016). The presence of the polar vortex isolates polar stratospheric air, and so the seasonal breakdown of the vortex allows sudden mixing and stirring of vortex air with ozone-rich mid-latitude air. The timing of the final warming modulates the strength and speed at which this

mixing occurs (Waugh and Rong, 2002).

    Just as for midwinter SSWs, changes in the stratosphere at the time of the final warming in spring can have an influence on weather patterns in both hemispheres (Black et al., 2006; Black and McDaniel, 2007a), including extreme events (Domeisen and Butler, 2020). In the SH, the tropospheric eddy-driven jet exhibits an equatorward shift at the time of the FSW related to the Southern Annular Mode (SAM) (Byrne et al., 2017; Byrne and Shepherd, 2018; Lim et al., 2018). The trend and variability

in the timing of the FSW event due to ozone depletion has been suggested to further affect the surface impact (Son et al., 2013). In the Northern Hemisphere, the FSW is associated with a weakening and equatorward shift of the North Atlantic storm track resembling the negative phase of the North Atlantic Oscillation (NAO), associated with high geopotential height anomalies over the Arctic (Black et al., 2006; Ayarzagüena and Serrano, 2009). Consistent with the chemical-dynamic feedbacks discussed above, spring ozone extremes have also been linked to anomalous surface weather patterns (Calvo et al., 2015; Ivy et al., 2017).





Furthermore, FSW events have been suggested to contribute to variability (Ayarzagüena and Serrano, 2009) and predictability (Byrne et al., 2019; Hardiman et al., 2011; Butler et al., 2019) at the surface. While SSWs cannot be predicted more than 1-2 weeks in advance (Taguchi, 2014, 2016; Karpechko et al., 2018; Karpechko, 2018), FSW events tend to be more predictable, especially events in late spring (Butler et al., 2019). The higher predictability of FSW events with respect to SSW events may provide enhanced lead times for potential surface impacts in comparison to SSW events. For a comprehensive comparison of

the predictability timescales of sudden and final stratospheric warming events see Domeisen et al. (2020).

SSW events have been classified according to a range of characteristics (Butler et al., 2015), notably with respect to the zonal wavenumber dominating the polar stratosphere at the time of or just prior to the event (Bancalá et al., 2012; Barriopedro and Calvo, 2014), or according to vortex elliptical moment diagnostics (Waugh, 1997; Charlton and Polvani, 2007; Mitchell et al., 2011; Seviour et al., 2013), that is, whether the vortex splits into two vortices or displaces off the pole. They have also been

classified with respect to their downward impact (Kodera et al., 2016; Runde et al., 2016; Karpechko et al., 2017; Charlton-Perez et al., 2018; Domeisen, 2019; Afargan-Gerstman and Domeisen, 2020). FSW events, on the other hand, have generally been classified according to the timing of their occurrence into "early" and "late" events (e.g. Waugh and Rong, 2002), and their altitude of origin in the stratosphere (Hardiman et al., 2011).

Planetary wave activity from the troposphere to the stratosphere is on average stronger in austral spring compared to austral

winter or boreal spring (Randel, 1988; Wang et al., 2019). Climatologically, in the SH late winter and spring the wave structure in the stratosphere is dominated by a quasi-stationary zonal wavenumber 1 (hereafter: wave-1) with contributions from a transient, eastward-moving zonal wavenumber 2 (hereafter: wave-2) (Randel, 1988; Mechoso et al., 1988; Manney et al., 1991; Waugh and Randel, 1999; Harvey et al., 2002; Ialongo et al., 2012), which may contribute to zonal asymmetries in ozone depletion (Kravchenko et al., 2012). In the NH, early FSW events tend to be predominantly wave-driven (e.g., Vargin

et al., 2020). In fact, there is no mechanistic difference between midwinter SSW events and early NH FSW events; they are merely differentiated through the evolution of the stratospheric winds after the event, as the definition of the SSW requires the winds after the event to return to westerly for a consecutive number of days (Charlton and Polvani, 2007). Late FSWs may also be partly wave-driven, although as the mean flow weakens in boreal spring due to changing solar radiation, less weakening by waves is required for an event to occur. Sun et al. (2011) show in a model study that FSW events tend to occur earlier if wave

driving is increased, and a correspondence has been found between the amplitude of wave-1 and the NH FSW date (Savenkova et al., 2012). Wave geometry can also be associated with nonlinear resonance of the vortex, a process suggested to be potentially important in SH spring (Scott, R and Haynes, P, 2002; Plumb, 2010). Given the timing of FSW events in spring when the polar vortex has already weakened, one could hypothesize that these events are more often caused by higher zonal wavenumbers (e.g. waves 2 and 3) as compared to wave-1, as these will be allowed to propagate into the weaker winds (Charney and Drazin,

1961; Matsuno, 1970; Plumb, 1989). Nonetheless, a classification of individual FSW events in the historical record based on geometrical wave structure, and the influence of the wave geometry on stratospheric ozone and surface impacts, does not yet exist.

This study explores the classification of FSW events by wave geometry (Sect. 2), the connections between wave geometry and dynamical behavior in the stratosphere (Sect. 3), ozone distribution (Sect. 4), and surface impacts (Sect. 5).



## 2 Detection and classification of FSW events


For consistency with the definition of midwinter SSWs, which is based on the reversal of the westerly winds at 10 hPa and 60° latitude (Charlton and Polvani, 2007), and because the winds at 50 hPa in the SH often do not have a clear reversal to summertime easterlies, here we define the date of the FSW at 10 hPa. FSW events are detected as the first date before June 30 (December 31) when the zonal mean zonal winds at 60° latitude and 10 hPa in the NH (SH) reverse to easterly and do not return

to westerly for more than 10 consecutive days (e.g. Butler and Gerber, 2018). Tables 1 and 2 list the calculated FSW dates using daily-mean data from ERA-interim reanalysis (Dee et al., 2011), which is available for the Jan 1979-Aug 2019 period. Note that – by chance – the FSW events occurred on the same date in consecutive years (1979 and 1980) in both hemispheres (i.e., this is not a typo). The median date of the final warming for the 1979-2019 ERA-interim record is April 12 in the NH and November 19 in the SH. Early events are those that occur more than 2 days prior to the median date and late events are

those that occur more than 2 days after the median date. In Fig. 1 and Table A1, we extend the detection of FSWs in the NH from 1958-1978 using JRA-55 reanalysis (Kobayashi et al., 2015). We do not examine FSWs in the SH prior to 1979 using JRA-55 because large-scale dynamical features related to stratosphere-troposphere coupling processes are not reliable due to lack of assimilated observations in the SH prior to satellite measurements (Gerber and Martineau, 2018). While the identified final warming dates during the shared period (1979-2019) of ERA-interim and JRA-55 can vary by 1-2 days, here we only

show JRA-55 FSW dates prior to 1979 and ERA-interim FSW dates after 1979.

We then classify FSW events by their geometry, either wave-1 or 2, according to three different methods at both 10 hPa and 50 hPa. The first two methods involve Fourier decomposition in the zonal direction of the 10 hPa or 50 hPa geopotential height anomalies averaged with cosine-weighting by latitude over 55-65° latitude. The anomalies are relative to the 1979-2018 climatology. We determine which wavenumber has, during the period from 10 days prior to 10 days after the FSW date, [1.]

the maximum amplitude for the greatest number of days, and [2.] the maximum mean amplitude (similar to Bancalá et al. (2012) and Barriopedro and Calvo (2014) for midwinter SSWs). The former measures the persistence and the latter indicates the strength of a given wavenumber; these frequently but not always yield the same classification.

Method [3.] involves the calculation of vortex elliptical moment diagnostics. We adapt the method from Seviour et al. (2013) which was applied to midwinter SSWs, using both the 10 hPa and 50 hPa geopotential height fields. To calculate the vortex edge

in NH (SH) spring, we use the 1979-2018 ERA-interim Feb-Apr (Sep-Nov) mean geopotential height value at 60° latitude. For the 10 days prior to and after the FSW, we calculate the aspect ratio and centroid latitude of the vortex. The aspect ratio is a measure of the elongation of the vortex, whereas the centroid latitude is a measure of how far the center of the vortex is displaced off the pole. A high aspect ratio (more elongated vortex) would indicate a "split" vortex while a low centroid latitude would indicate a "displaced" vortex. For the NH at 10 (50) hPa, we find the number of days where the aspect ratio exceeds a

value of 2.20 (2.25) or where the centroid latitude is below a value of 64 (68)°N. For the SH at 10 (50) hPa, we find the number of days where the aspect ratio exceeds a value of 1.80 (1.70) or where the centroid latitude is below a value of 71 (78)°S. These thresholds are based on the 10th percentile of climatological FMA (SON) values in the ERA-interim record for the NH (SH). If the number of days exceeding the centroid latitude threshold is greater (smaller) than the number of days exceeding





**Table 1.** Dates and classifications for all FSW events for the Northern and Southern Hemisphere according to ERA-interim data. Early (late) events are indicated in **bold** (*cursive*), referring to a date before (after) the median date of April 12 in the NH and November 19 in the SH. Dates that fall within ± 2 days of the median date are not classified as early or late. U = unclassified (methods did not agree according to the criterion outlined in section 2). Superscripts indicate the JRA-55 classification if it was not in agreement with ERA-interim.

| year | NH FSW date | wave @ 10hPa | wave @ 50hPa | SH FSW date | wave @ 10hPa | wave @ 50hPa |
|---|---|---|---|---|---|---|
| 1979 | **Apr-8** | wave-1 | wave-1 | Nov-18 | wave-1 | wave-2 |
| 1980 | **Apr-8** | wave-2 | U | Nov-18 | wave-1 | wave-1 |
| 1981 | *May-12* | U | wave-2[U] | Nov-17 | wave-2 | wave-2 |
| 1982 | **Apr-4** | wave-1 | wave-1 | Nov-19 | wave-1 | wave-2 |
| 1983 | **Apr-1** | wave-1 | wave-1 | **Nov-8** | wave-1[2] | wave-2 |
| 1984 | *Apr-25* | wave-1 | wave-2 | **Nov-6** | wave-2[U] | wave-2 |
| 1985 | **Mar-24** | wave-1 | wave-1[U] | *Nov-25* | wave-1 | wave-1 |
| 1986 | **Mar-20** | wave-1 | wave-1 | **Nov-13** | wave-1 | wave-1 |
| 1987 | *May-5* | wave-1 | U | *Dec-2* | wave-1 | wave-1 |
| 1988 | **Apr-6** | wave-1 | wave-1 | **Oct-27** | wave-1 | wave-1 |
| 1989 | *Apr-15* | wave-2 | wave-1 | **Nov-11** | wave-1 | wave-1 |
| 1990 | *May-8* | U[1] | wave-2 | *Dec-5* | wave-1 | wave-1 |
| 1991 | Apr-10 | wave-1 | wave-1 | **Nov-15** | wave-2 | wave-2 |
| 1992 | **Mar-22** | wave-1 | wave-1 | Nov-21 | wave-1 | wave-1 |
| 1993 | Apr-12 | wave-1 | wave-1 | *Nov-23* | wave-1 | wave-2 |
| 1994 | **Apr-2** | wave-2 | wave-2 | **Nov-12** | wave-1 | wave-1 |
| 1995 | **Apr-8** | wave-1 | wave-1 | *Nov-24* | wave-1 | wave-1 |
| 1996 | Apr-10 | wave-1 | wave-1 | *Dec-4* | wave-1 | wave-1 |
| 1997 | *Apr-30* | wave-1 | wave-1 | Nov-18 | wave-1 | wave-1 |
| 1998 | **Mar-29** | wave-1 | wave-1 | *Dec-7* | wave-1 | wave-2 |
| 1999 | *May-3* | wave-1 | wave-1 | *Dec-5* | wave-1 | wave-1 |

the aspect ratio threshold, the event is classified as a wave-1 (wave-2) event. Here, we make the assumption that a "split" event

can be classified as having primarily wave-2 structure (and "displacement" events as having wave-1 structure) but note that some "split" midwinter SSWs are preceded predominantly by wave-1 activity, especially at long lead times (Labitzke, 1981; Bancalá et al., 2012). Another disadvantage of this method is that, as the sharp edge of the vortex (marked by a strong potential vorticity gradient) starts to weaken and dissipate in spring, the elliptical moments cannot be diagnosed, so this method is unable to classify FSW events as consistently as the other two methods.

For every event, each of these three methods indicates a preference for either wave-1 or wave-2, or it returns an "unclassified" result. The final wave geometry classification used throughout the remainder of this study is then determined based on the agreement of at least two of the above methods. If no two methods agree, or if two methods yield a non-classification, the event





**Table 2.** Continuation from Table 1.

| year | NH FSW date | wave @ 10hPa | wave @ 50hPa | SH FSW date | wave @ 10hPa | wave @ 50hPa |
|------|-------------|--------------|--------------|-------------|--------------|--------------|
| 2000 | Apr-10 | wave-1 | wave-1 | **Nov-4** | wave-1 | wave-1 |
| 2001 | *May-10* | wave-1 | wave-1 | *Dec-7* | wave-1 | wave-2 |
| 2002 | *May-4* | wave-1 | wave-2 | **Nov-1** | wave-1 | wave-1 |
| 2003 | Apr-14 | wave-1 | wave-2 | **Nov-15** | wave-1 | wave-1 |
| 2004 | *Apr-30* | wave-1 | wave-2 | Nov-16 | wave-2$^U$ | wave-2 |
| 2005 | **Mar-13** | wave-2[1] | wave-1 | **Nov-10** | wave-1 | wave-1 |
| 2006 | *May-7* | wave-1 | wave-1 | *Dec-3* | wave-1 | wave-2 |
| 2007 | *Apr-19* | wave-1 | wave-1 | *Nov-27* | wave-1 | wave-1 |
| 2008 | *May-1* | wave-1 | wave-2[1] | *Dec-1* | wave-1 | wave-1 |
| 2009 | *May-10* | wave-2 | wave-2 | Nov-16 | wave-1 | wave-1 |
| 2010 | *Apr-30* | wave-1 | wave-1 | *Dec-11* | wave-1 | wave-1 |
| 2011 | **Apr-5** | wave-1 | wave-1 | *Nov-25* | wave-1 | wave-1 |
| 2012 | *Apr-19* | wave-1 | wave-2 | **Nov-6** | wave-1 | wave-2 |
| 2013 | *May-4* | wave-1 | wave-1 | **Nov-2** | wave-1 | wave-1 |
| 2014 | **Mar-27** | wave-1 | wave-2 | *Nov-22* | wave-1 | wave-1 |
| 2015 | **Mar-28** | wave-1 | wave-1 | *Dec-11* | wave-1 | wave-1 |
| 2016 | **Mar-5** | wave-1 | wave-1 | **Nov-10** | wave-2$^U$ | wave-1 |
| 2017 | **Apr-8** | wave-1 | wave-1 | **Nov-9** | wave-1 | wave-1 |
| 2018 | *Apr-15* | wave-1 | wave-1 | *Nov-24* | wave-1 | wave-1 |
| 2019 | *Apr-23* | wave-1 | wave-1 | | | |

is labeled as "unclassified". Table A2 shows the individual classification at 50 hPa for each of the three methods for ERA-interim, as a demonstration of how the final classification was determined. Tables 1 and 2 show the final classification at 10 and
50 hPa for the satellite era for both hemispheres, and Table A1 indicates the classification for the pre-satellite era, i.e. 1958 - 1978, for the NH. For the period 1979 - 2019, we check the classifications using both ERA-interim and JRA-55 reanalysis data, as wave geometry for midwinter SSWs has been found to be sensitive to the reanalysis used (Gerber et al., 2021). In general, the classification of FSW events is consistent across the two reanalysis products, although a few discrepancies are noted in Tables 1 and 2.

Figure 1a,c illustrates the sequence of dates of the final warmings along with their wave geometry classification, and their timing of occurrence with respect to the median final warming date, indicated by a horizontal dashed line. In general, there are far fewer wave-2 events compared to wave-1 events in the 41-year record since 1979. Using the 10 hPa classification, there are 5 wave-2 events each in the NH and SH. Using the 50 hPa classification, roughly a quarter of final warmings have wave-2 structure; there are 11 (12) wave-2 events in the NH (SH) during this period. This frequency of wave-2 events is slightly

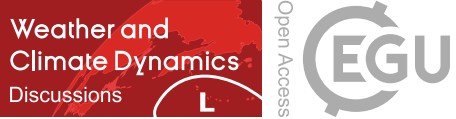

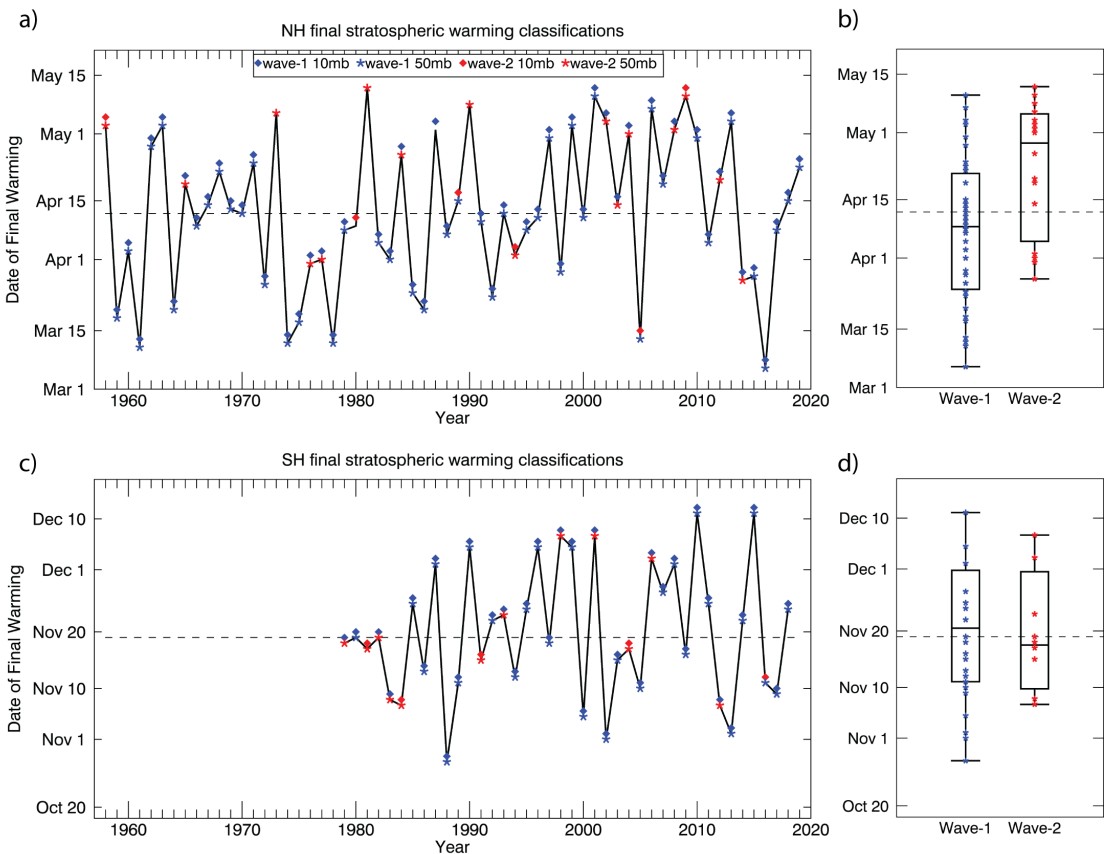

**Figure 1.** (a,c) Dates of FSWs in the NH and SH, using JRA-55 reanalysis for 1958-1978 and ERA-interim reanalysis for 1979-2019. Symbols indicate the classification of the event at both 10 hPa and 50 hPa using ERA-interim. (b,d) Dates of FSWs grouped by either wave-1 or wave-2 classification at 50 hPa. The whiskers show the earliest/latest dates, the top/bottom of the box shows the upper and lower quartiles, and the solid line shows the median date for each classification. The horizontal dashed lines indicate the median date (based on the 1979-2019 period) for all final warmings in each hemisphere.

larger than the frequency of wave-2 midwinter SSWs (e.g., Barriopedro and Calvo (2014), who found 9 wave-2 events in the 1958-2010 period, using a 50 hPa geopotential height Fourier decomposition method).

Based on the classification at 50 hPa, in the NH, wave-2 events occur significantly later than wave-1 events (Fig. 1b), with 11 out of 16 wave-2 events from 1958-2019 occurring at least 2 days later than the median date of April 12. In the SH, no statistical difference between the date of wave-1 and wave-2 FSW events is observed from 1979-2018 (Fig. 1d). As noted

previously, the SH final warming date tends to occur later during years of strong ozone depletion, as the late breakup is linked via chemistry-climate feedbacks to stronger ozone loss that further cools and strengthens the vortex into late spring.

For illustration, Fig. 2a-h shows selected wave-1 and wave-2 cases of FSWs. Different years were selected for each level in order to showcase the presence of wave structures throughout the record during the satellite era. The wave-1 and wave-2





events show geopotential height structures that are strongly reminiscent of the structures observed during wave-1 and wave-
2 midwinter SSW events, with the vortex shifted off the pole during wave-1 events and elongated or split into two smaller
vortices during wave-2 events. Although no event showed a dominant wave-3 pattern according to the above classifications,
visual inspection and quantification of the wave-3 component using the Fourier decomposition method reveal a substantial role
of wave-3 in some cases, which are highlighted in Fig. 2i-l.

Further evidence that wave-3 plays a more significant role in NH FSW events compared to midwinter SSW events is provided
by comparing the ratio of wave-2 and wave-3 amplitudes to wave-1 amplitude averaged for the 15 days prior to SSW and FSW
events (Fig. 3). For both SSWs and FSWs, wave-2 and wave-3 amplitudes tend to be more comparable to wave-1 amplitudes in
the troposphere (200 hPa); while in the lower stratosphere (50 hPa), wave-2 and 3 typically have smaller amplitudes than wave-
1 (indicated by median ratios less than one), as expected from wave filtering (Charney and Drazin, 1961). However, prior to
SSWs, wave-2 amplitudes are sometimes much larger than wave-1 amplitudes. The large spread, which is skewed toward more
positive ratios of wave-2 relative to wave-1 prior to SSWs at all pressure levels, may be an indication of the role of non-linear
resonance just prior to these events (Esler and Matthewman, 2011; Domeisen et al., 2018; Albers and Birner, 2014). Wave-3
amplitudes are generally much smaller relative to wave-1 and wave-2 prior to SSWs. This is true for FSWs as well; however,
the median ratios of both wave-2 and wave-3 relative to wave-1 for FSWs are higher than for SSWs at all levels (particularly
at 50 hPa), suggesting that wave-2 and wave-3 are able to propagate higher as the westerly flow weakens in spring.

## 3  Relationship between geometry and dynamical behavior

In this section we investigate the stratospheric dynamical characteristics of the final warming events. Figure 4 shows a com-
posite of zonal mean zonal wind around the time of the FSW using the date of the FSW at 10 hPa and the wave classification
at the depicted level, i.e. 10 hPa for Figs. 4a,c and 50 hPa for b,d. The wind speeds about a month before the FSW event at
10 hPa are weaker in the NH as compared to the SH (Figs. 4a,c). In the NH the winds can already exhibit values close to zero
within the month before the FSW event, while in the SH the winds are significantly stronger in the month before the event. The
average deceleration between lags of -30 to -11 days before the FSW event and days 11 to 30 after the event is 22.7 m/s (35.6
m/s) for the NH (SH) at 10 hPa. Further down at 50 hPa, the deceleration is smaller, i.e. 12.6 m/s (30.4 m/s) for the NH (SH).

When comparing the rate of wind deceleration with respect to wave geometry, wave-2 events in the SH (red lines in Fig. 4c,d)
exhibit a stronger deceleration as compared to wave-1 events. The average deceleration between lags of -30 to -11 days before
the FSW event and days 11 to 30 after the event is 35.0 m/s (40.5 m/s) for wave-1 (wave-2) at 10 hPa and 29.8 m/s (34.2 m/s)
at 50hPa (Fig. 4c,d). For the NH, this difference is less distinct. In the NH the average deceleration over the same period is
23.5 m/s (20.8 m/s) for wave-1 (wave-2) events at 10 hPa, and 12.8 m/s (12.3 m/s) for wave-1 (wave-2) events at 50 hPa. The
deceleration associated with wave-2 is greater than for wave-1 events only for the $\pm$ 10 days around the FSW event (Fig. 4a,b).
Note the larger variability for wave-2 events due to the smaller sample size. Within the month after the event occurs at 10 hPa,
no significant difference in wind speed can be found between wave-1 and wave-2 events at 10 or 50 hPa for either hemisphere.



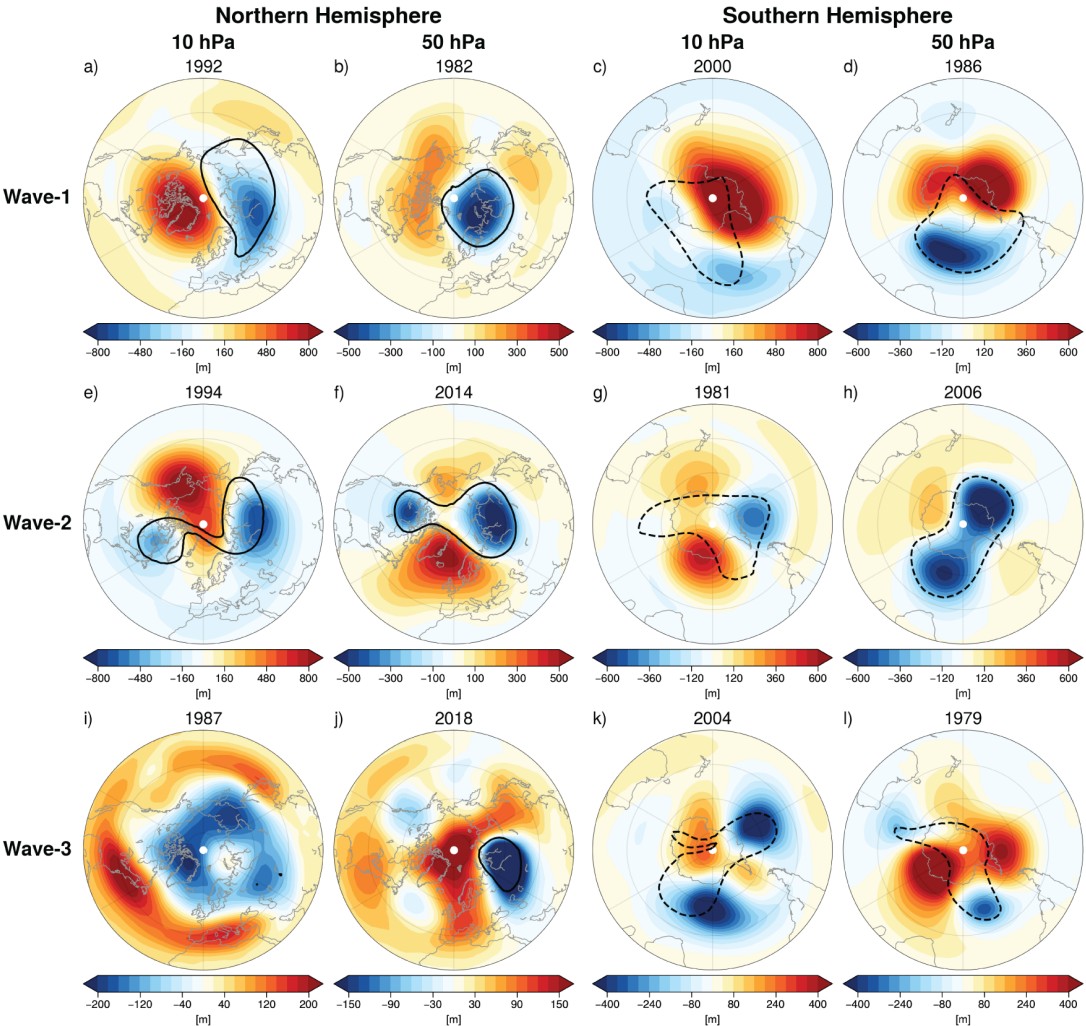

**Figure 2.** Geopotential height anomalies (shading) and potential vorticity at the 500K isentrope (solid contour in NH or dashed contour in SH) from ERA-interim reanalysis on the day of the final warming for selected case studies that show a clear wave structure for (a-d) wave-1, (e-h) wave-2, and (i-l) wave-3, for both hemispheres, and for both 10 and 50hPa. Note the different colorbars. The PV contour is the 40 PVU (-40 PVU) value in the NH (SH) and gives a qualitative estimate for the location and shape of the polar vortex.

Early FSW events are associated with a stronger deceleration of the winds as compared to late FSW events due to the seasonally stronger winds earlier in the season (not shown). At 50 hPa, winds are significantly weaker around the time of the FSW event in the NH as compared to the SH, and in the NH the winds often cross zero on average within 1-2 weeks of the time of the FSW at 10 hPa. In the SH the winds remain significantly stronger at 50 hPa even though the winds reverse at 10hPa, and only a few events turn to easterly zonal mean zonal wind speed at 50 hPa within the month after the FSW event is detected at 10 hPa. On average, on the day of the reversal to easterlies at the 10 hPa level the zonal mean zonal winds at 60° latitude

195





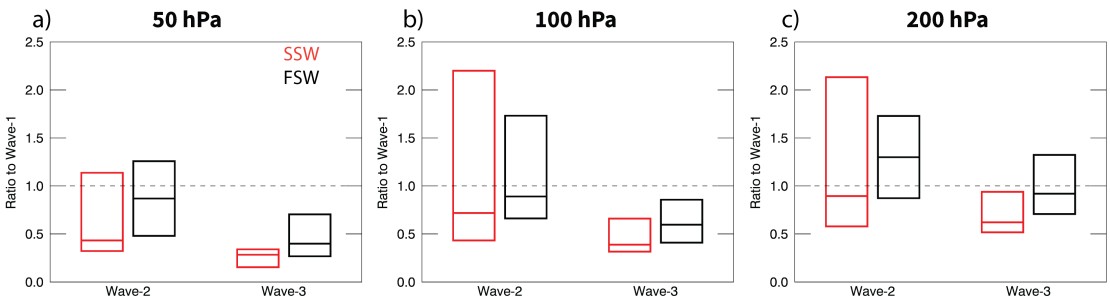

**Figure 3.** Ratio of wave-2 and wave-3 amplitudes relative to wave-1 amplitudes, averaged for the 15 days prior to either midwinter SSW events (red) or FSW events (black) for the 1979-2019 period in the NH, at (a) 50 hPa, (b) 100 hPa, and (c) 200 hPa. The top/bottom of the boxes show the quartile range and the solid horizontal line shows the median value for 23 midwinter SSW events and 41 FSW events. The dashed line shows where the ratio of amplitudes is equal to 1. The midwinter SSW dates are from Butler et al. (2017).

and 50 hPa are at 5.8 m/s (19.2 m/s) for the NH (SH). Given the different classifications for FSW events in the literature, it is important to note that detecting the FSW at 10 hPa or 50 hPa based on the wind reversal to easterlies yields a much more significant shift in the timing of the SH as compared to the NH (Newman, 1986).

## 4    Implications for ozone distribution during spring onset

To investigate the influence of final warming wave geometry on total column ozone, we use the Bodeker Scientific Filled Total Column Ozone (TCO) Database (Version 3.4) (Bodeker et al., 2020). This dataset combines measurements from multiple satellite-based instruments, and fills missing data with a machine-learning based method to create a temporally and spatially gap-free database of total column ozone from 31 Oct 1978 to 31 Dec 2016. We also compared these results to the same analysis using ERA-interim ozone at the 500K isentrope (lower stratosphere) and the results were very similar (not shown). TCO anomalies are calculated based on the 1979-2016 daily climatology. Here we composite TCO anomalies based on FSW dates at the 10 hPa level but using classifications at 50 hPa, near where ozone levels peak in the polar stratosphere.

Figure 5a-h shows the TCO anomaly for wave-1 and wave-2 FSW events 15 days prior to and after the final warming in both hemispheres (from 1979-2016, the period of the ozone dataset). In the NH, negative TCO anomalies are present inside the polar vortex region over the 15 days before the final warming for both wave-1 and wave-2 events, though the negative anomalies extend further over Eurasia and the North Atlantic prior to wave-1 events, when the vortex tends to be more displaced compared to wave-2 events. Positive TCO anomalies are found over the polar cap after the final warming for both wave-1 and wave-2 events, due to the sudden increase in ozone as the vortex dissipates and mixing with ozone-rich mid-latitude air can occur. Similarly, mid-latitude regions experience negative TCO anomalies after the final warming; de la Cámara et al. (2018) attribute a similar feature after midwinter SSWs to cross-isentropic advection and isentropic irreversible mixing. Although there are

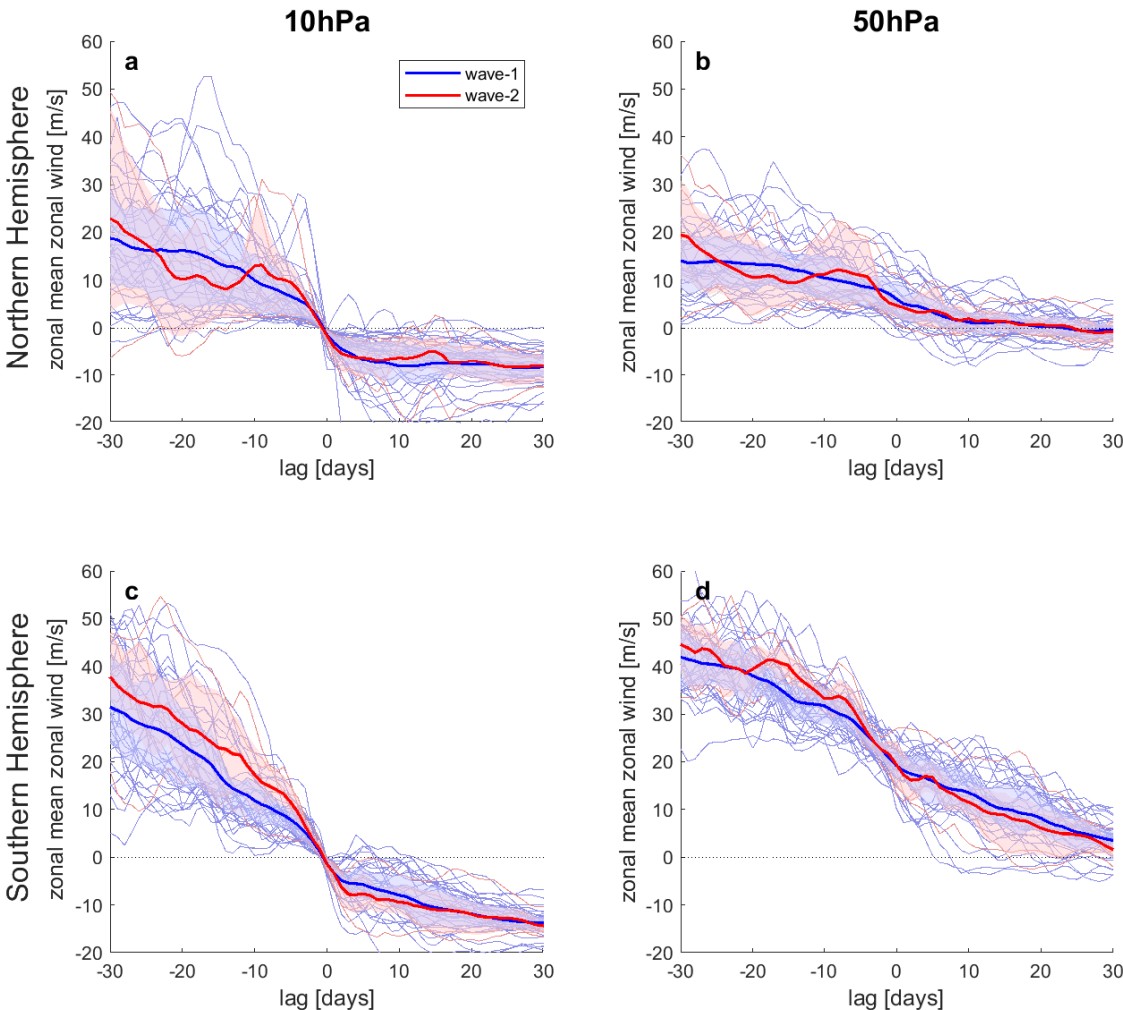

**Figure 4.** Composites of zonal mean zonal wind [m/s] at 60° latitude and 10 hPa (a,c) and 50 hPa (b,d) for the NH (a,b) and the SH (c,d) for 1979 - 2019 (NH) and 1979 - 2018 (SH) for lags of -30 to +30 days around all final warming event dates defined at 10 hPa according to the ERA-interim classification. Thin blue (red) lines correspond to a wave-1 (wave-2) classification at the depicted pressure level. Thin gray lines (if applicable) correspond to FSW events unclassified by wavenumber at the depicted level. The bold blue (red) lines indicate the average of all events classified as wave-1 (wave-2) at the depicted level. The blue (red) shading indicates the area between the 25th and 75th percentile for the wave-1 (wave-2) classification.

some qualitative differences between wave-1 and wave-2 TCO anomalies in the NH, the significance levels suggest that the differences between wave-1 and wave-2 events are few.

In the SH, TCO anomalies associated with wave-1 events mirror wave-1 events in the NH, with a displaced region of negative TCO anomalies over the Antarctic peninsula prior to the FSW, and positive TCO anomalies over the polar cap and negative

**Figure 5.** Composite total column ozone anomaly [Dobson Units] for (a-h) wave-1 vs wave-2 events, and (i-p) early vs late events, ± 15 days around the final warming date (using the wave classification method at 50 hPa in both hemispheres). Dots show regions that are significantly different between the respective panels (i.e., (a-h) wave-1 vs wave-2, and (i-p) early vs late) at the 95% level.

TCO anomalies in the mid-latitudes after the FSW. The TCO anomalies for wave-2 events in the SH have a more unique structure, with a broad ring of positive TCO anomalies over the Southern Ocean prior to the FSW, where the anomalies over





the Drake passage/Amundsen Sea are significantly different from wave-1 events. However, after the FSW, wave-2 events are also followed by enhanced TCO over the polar cap.

Though the wave geometry does indicate qualitative influences on the spatial structure of TCO anomalies, we find a more
substantial role of the timing of the FSW on the TCO anomalies, particularly in the SH (Fig. 5i-p). In the NH, where wave-1 events tend to occur earlier and wave-2 events tend to occur later (Fig. 1b), the TCO anomalies for early and late events are qualitatively similar to the wave-1 and wave-2 composites (Fig. 5a,b,e,f), except that the positive TCO anomalies over North America prior to, and over the polar cap after, early FSWs are larger and statistically different from the anomalies during late FSWs. In the SH, however, early FSWs show positive TCO anomalies both prior to and after the event, while late FSWs show
negative TCO anomalies both prior to and after the event (though for both early and late cases, the TCO anomaly is more positive after the event than before). These differences reflect a key point, which is that late events in the SH tend to occur in years with strong ozone depletion that further strengthen the vortex winds and allow the vortex to persist longer. Moreover, because the SH vortex in the lower stratosphere persists for several weeks after the FSW occurs at 10 hPa (Fig. 4), there appears to be less mixing of polar vortex and mid-latitude ozone in the 15 days after the FSW, compared to the NH (Newman, 1986).

As found in previous studies, the FSW is clearly important in the evolution of springtime TCO in both hemispheres, with the event leading to more positive TCO anomalies over the polar cap no matter the classification. We further show that both the wave geometry and timing of the event can play a role in the springtime TCO anomalies, which may have implications for ecosystems and human health due to increased ultraviolet (UV) radiation exposure (Barnes et al., 2019). For example, prior to wave-1 (and early NH) FSWs there are more widespread negative TCO anomalies that are shifted further off the pole towards
more populated areas (Fig. 5a,c,i). Late events in the NH induce much weaker changes in the TCO anomalies, whereas late events in the SH are associated with persistent negative polar cap TCO anomalies that reflect chemistry-climate feedbacks due to large ozone depletion events.

## 5   Surface impacts

We next investigate potential differences in the surface impact between wave-1 and wave-2 FSW events, using the classification
at 50 hPa. The surface structures using the classification at 10 hPa (not shown) do not differ qualitatively from the surface impacts based on the 50 hPa classification. Figure 6 shows the composite response for wave-1 and wave-2 FSWs for linearly detrended 500hPa geopotential height anomalies for 60 days after the FSW event. The detrending was applied to account for possible trends in the storm tracks but does not qualitatively change the results. The average over all NH FSW events (Fig. 6a) shows a negative NAO structure with a high pressure anomaly over Greenland and a low pressure anomaly over Europe and
the adjacent North Atlantic region. A negative geopotential height anomaly is observed in the North Pacific off the coast of western Canada. When dividing the response between wave-1 and wave-2 (Fig. 6c,e), the negative NAO response persists for both types of events, but the response over North America is opposite between wave-1 and wave-2 events, with a positive (negative) anomaly over Canada for wave-1 (wave-2) events and the opposite response over the southern U.S. These differences between wave-1 and wave-2 events over North America are significant. Other areas of significant differences between wave-1





and wave-2 events are small areas in the Arctic ocean and the southern North Pacific. Worth noting is that observed differences in the NH surface impacts in the weeks following displacement and split-type SSW events include increased blocking over Canada for displacement events and a stronger negative NAO pattern for split SSWs (Mitchell et al., 2013). These general patterns are echoed here for wave-1 and wave-2 FSW events, respectively. However, these surface responses are strongly limited by sampling, as the number of events, especially for wave-2, is small (Maycock and Hitchcock, 2015).

In the Southern Hemisphere, a dominant high pressure region is found in the Amundsen Sea in the average for all events (Fig. 6b). Separating the events into wave-1 and wave-2 (Fig. 6d,f) reveals that this positive anomaly is dominantly contributed by wave-1 events, while wave-2 events exhibit a significantly opposite anomaly in the same region. Other areas of significantly opposite responses between wave-1 and wave-2 lie east of Drake passage, south of the African continent, and near New Zealand.

Since the surface signal over the North Atlantic tends to show a structure reminiscent of the negative phase of the NAO (Fig. 6a), we also composite the NAO index (obtained from the Climate Prediction Center) for the period 1979 - 2019 using the wavenumber classification from ERA-interim at 50 hPa (Fig. 7). The NAO experiences a decrease from significantly positive values before the FSW event to values close to zero or negative within the 60 days after the event. The results do not change qualitatively when using the JRA-55 wave classification for FSWs (1958 - 2019). Both wave-1 and wave-2 FSW events

experience a tendency towards a negative NAO after the central day of the event, with on average consistently positive NAO values in the two months before the event. Values significantly different from zero are observed primarily for wave-1 events for lags between 15 to 40 days before the FSW event. Wave-2 events show larger variability, likely due to the smaller sample size as compared to wave-1 events.

## 6  Conclusions

Both sudden stratospheric warming events in the middle of winter as well as final stratospheric warming events that mark the end of winter in the stratosphere are characterized by a similar evolution and are often classified by the same thresholds, i.e. a reversal of the zonal mean zonal winds of the polar vortex. However, in order to characterize their evolution further, different measures are used. The most dominant classification for midwinter sudden stratospheric warmings is by their wave geometry into split and displacement events. FSWs, on the other hand, have so far not been classified by geometry, but only

by their timing or vertical evolution. This difference in the classification between midwinter and end-of-winter polar vortex breakdowns is likely due to the notion that a wave geometry cannot always be identified for FSW events, especially for events that occur later in spring and that are more radiatively driven. We show here that final warmings can almost exclusively be classified with regard to their geometrical wave structure. This geometrical structure is present for all but two years in the NH (at both 10 hPa and 50 hPa) and notably even for most late events. A detailed classification according to several methods and

datasets is provided in the manuscript.

In the NH, wave-1 events tend to occur earlier and wave-2 events tends to occur later, which may be surprising considering early NH events are more dynamically-forced; on the other hand, weaker westerly winds in spring allow for more vertical

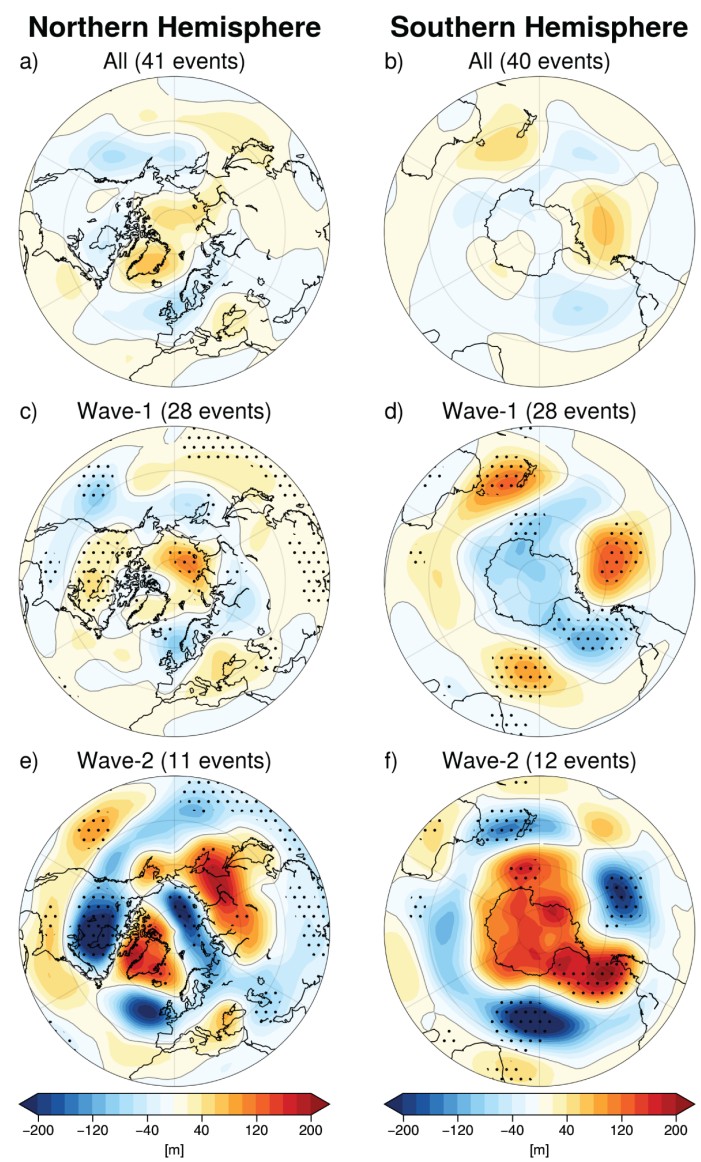

**Figure 6.** Composite of the linearly detrended 500hPa geopotential height anomalies [m] from ERA-interim data for (a,b) all FSW events, and FSW events classified according to ERA-interim as (c,d) wave-1 or (e,f) wave-2 at 50 hPa averaged over the 60 days after the central FSW date (i.e., lags of 1-60 days) for (a,c,e) the NH and (b,d,f) the SH extratropics. The stippling indicates regions where the wave-1 and wave-2 composites are significantly different from each other at the 95% confidence level according to a 1000-sample bootstrap analysis (with replacement).

propagation of wave-2 and even wave-3 into the stratosphere. Similar to SSWs, more events are characterized as wave-1 events as compared to wave-2 events in both hemispheres. Wave-3 plays a more significant role in the NH stratosphere during the



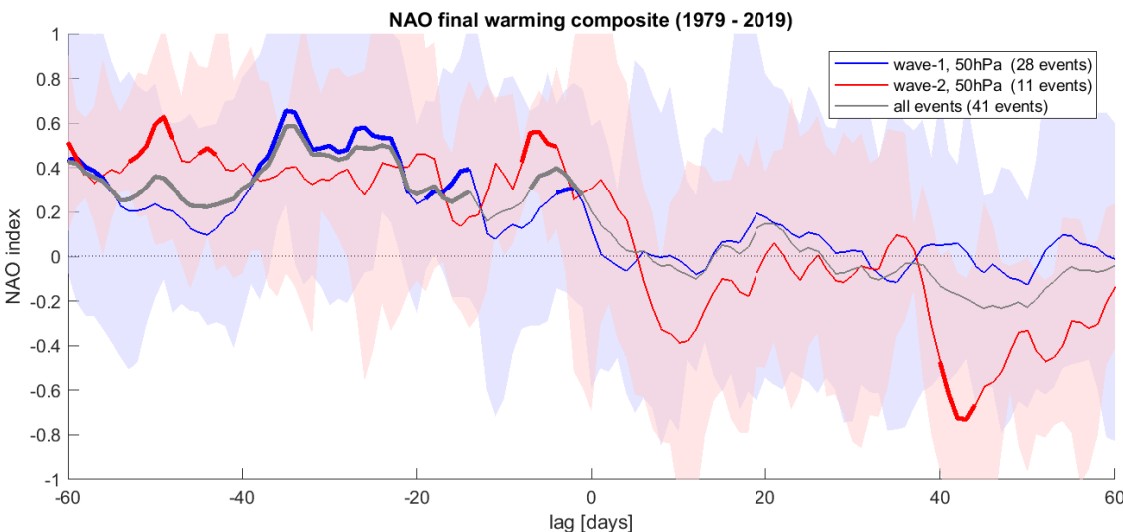

**Figure 7.** Composite of the NAO index for lags of -60 to +60 days around the final warming central date using the wavenumber classification at 50 hPa based on ERA-interim data for 1979 - 2019. The blue (red) lines indicate the average values for the wave-1 (wave-2) classifications at 50 hPa, respectively. The gray line is the average over all FSW events from 1979 - 2019, including wave-1, wave-2, as well as unclassified events. Bold parts of the lines indicate values significantly different from zero at the 5% level according to a t-test. The blue (red) shading indicates the 25th and 75th percentiles for the wave-1 (wave-2) classification.

FSW compared to midwinter SSW events, when wave-3 is generally not able to propagate into the strong vortex winds present prior to SSWs. Nevertheless, wave-1 and wave-2 still tend to dominate during FSW events; none of the identified FSWs was classified as a wave-3 event according to the classification methods used in this study.

In the NH, the winds at 60°N and 50 hPa turn to easterlies on average about 10-15 days after they turn to easterlies at 10 hPa, while in the SH this transition between 10 and 50 hPa takes more than 30 days. Hence classifying the final warming at 10
hPa or 50 hPa based on the wind reversal to easterlies yields a much more significant shift in the timing in the SH as compared to the NH. Indeed, in the NH picking 10 or 50 hPa does not yield a strong difference for the surface impact, at least for wave-1, where there is a sufficient number of cases to average over.

Stratospheric ozone is substantially modified by the final warming, with generally ozone-poor air within the vortex prior to the event, which then mixes with ozone-rich mid-latitude air after the event as the strong barrier associated with the polar
vortex weakens. While the timing of the FSW plays a dominant role in the overall strength of the total column ozone anomalies before and after the event in both hemispheres, the wave geometry can have some influence. In particular, wave-1 events tend to be associated with more negative and widespread TCO anomalies prior to the FSW than wave-2 events in both hemispheres. This may be because, while wave-2 events tend to be associated with elongation and possible splitting over the pole, wave-1 events displace the vortex equatorward into more sunlit regions. In the SH, differences in polar cap TCO anomalies for early
and late events are particularly noticeable, and are likely associated with chemistry-climate feedbacks that play a central role





in stratosphere-troposphere coupling in austral spring. Consideration of the timing of the FSW in both hemispheres may be important for how much stratospheric ozone is transported via deep stratospheric intrusions to the surface in spring (Albers et al., 2018; Breeden et al., 2020).

Similar to SSW events, NH FSWs tend to yield a positive geopotential height anomaly at 500 hPa over Greenland for the 60 days following the final warming. Consistent with this the surface NAO decreases from before the event to after the event, no matter the classification. The geopotential height anomaly response over North America is the opposite for wave-1 and wave-2 events, with a more positive (negative) height anomaly over Canada for wave-1 (wave-2) events. In the SH, the response is also opposite for wave-1 and wave-2 events, with negative (positive) anomalies over the polar cap for wave-1 (wave-2) events. We did not attempt to identify causes for the different surface responses, which could be linked to tropospheric variability leading up to the FSWs, the more direct influence of the stratospheric wave geometry on the underlying tropospheric circulation, or to other large-scale climate patterns like El Niño-Southern Oscillation (Domeisen et al., 2019) or decadal variability. These opposing signals in both hemispheres may also arise due to sampling, given the small number of events available in the historical record; further testing with long model simulations may reveal non-significant differences (e.g. Maycock and Hitchcock, 2015). Furthermore, the differences in the SH surface impact between wave-1 and wave-2 events could potentially be related to the trends associated with ozone depletion and recovery and will have to be investigated further.

The ability to classify final stratospheric warming events by wave geometry points out similarities with midwinter sudden stratospheric warming events, while the greater importance of wave-3 for FSWs highlights the differences. We have shown that the structure of the stratospheric polar vortex as it weakens in spring can influence total column ozone and tropospheric impacts, suggesting that the wave geometry of FSWs may be important for improving predictive skill following these events. Whether the wave geometry characteristics of FSWs are well simulated in climate and forecast models, and if they are modulated by external forcings like increasing greenhouse gases, should be investigated.

*Data availability.* The ERA-interim Reanalysis data was obtained from the ECMWF data portal at https://apps.ecmwf.int/datasets/data/interim-full-daily/. The JRA-55 data was obtained from the NCAR Research Data Archive at https://rda.ucar.edu/datasets/ds628.0/. The NAO index was obtained from the Climate Prediction Center at https://www.cpc.ncep.noaa.gov/products/precip/CWlink/pna/nao.shtml. The total column ozone database was obtained from https://zenodo.org/record/3908787#.X9JdLV57ns1. We would like to thank Bodeker Scientific, funded by the New Zealand Deep South National Science Challenge, for providing the combined NIWA-BS total column ozone database.

**Appendix A**



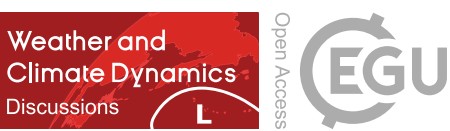

**Table A1.** Extension of Tables 1 and 2 for NH FSW dates from 1958 - 1978 based on JRA-55 reanalysis data.

| year | NH FSW date | wave @ 10hPa | wave @ 50hPa |
|------|-------------|--------------|--------------|
| 1958 | *May-3* | wave-2 | wave-2 |
| 1959 | **Mar-18** | wave-1 | wave-1 |
| 1960 | **Apr-2** | wave-1 | wave-1 |
| 1961 | **Mar-11** | wave-1 | wave-1 |
| 1962 | *Apr-28* | wave-1 | wave-1 |
| 1963 | *May-3* | wave-1 | wave-1 |
| 1964 | **Mar-19** | wave-1 | wave-1 |
| 1965 | *Apr-19* | wave-1 | wave-2 |
| 1966 | **Apr-9** | wave-1 | wave-1 |
| 1967 | **Apr-14** | wave-1 | wave-1 |
| 1968 | *Apr-21* | wave-1 | wave-1 |
| 1969 | **Apr-13** | wave-1 | wave-1 |
| 1970 | **Apr-12** | wave-1 | wave-1 |
| 1971 | *Apr-24* | wave-1 | wave-1 |
| 1972 | **Mar-25** | wave-1 | wave-1 |
| 1973 | *May-6* | U | wave-2 |
| 1974 | **Mar-12** | wave-1 | wave-1 |
| 1975 | **Mar-17** | wave-1 | wave-1 |
| 1976 | **Mar-30** | wave-1 | wave-2 |
| 1977 | **Apr-1** | wave-1 | wave-2 |
| 1978 | **Mar-12** | wave-1 | wave-1 |



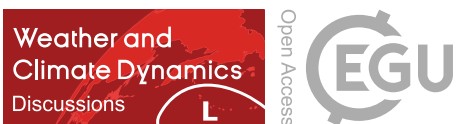

**Table A2.** Details of ERA-interim classification at 50 hPa, using final warming dates based on 60N/S and 10 hPa. Method 1 finds either a split (S), displacement (D), or unclassified (U) result, while Method 2 and 3 evaluate the number of days of greatest amplitude and maximum mean amplitude for wavenumbers 1-3 for the ± 10 days before and after the FSW, as described in Section 2.

| NH FW dates | # of days (Displacement, Split) | WN 1,2,3 % days | WN with greatest mean amplitude | Final classification | SH FW dates | # of days (Displacement, Split) | WN 1,2,3 % days | WN with greatest mean amplitude | Final classification |
|---|---|---|---|---|---|---|---|---|---|
| 8-Apr-79 | S (4,10) | 76, 24, 0 | 1 | wave 1 | 18-Nov-79 | U (5,5) | 10, 71, 19 | 2 | wave 2 |
| 8-Apr-80 | U (6,6) | 52, 43, 5 | 2 | U | 18-Nov-80 | D (8,4) | 43, 24, 33 | 1 | wave 1 |
| 12-May-81 | U (0,0) | 33, 48, 19 | 2 | wave 2 | 17-Nov-81 | S (7,10) | 19, 67, 14 | 2 | wave 2 |
| 5-Apr-82 | D (1,0) | 100, 0, 0 | 1 | wave 1 | 19-Nov-82 | S (2,7) | 33, 52, 14 | 2 | wave 2 |
| 1-Apr-83 | D (13,5) | 71, 0, 29 | 1 | wave 1 | 8-Nov-83 | S (0,7) | 5, 81, 10 | 2 | wave 2 |
| 25-Apr-84 | D (1,0) | 33, 52, 14 | 2 | wave 2 | 6-Nov-84 | S (3,4) | 38, 57, 5 | 2 | wave 2 |
| 24-Mar-85 | U (0,0) | 52, 29, 19 | 1 | wave 1 | 25-Nov-85 | D (9,4) | 86, 5, 10 | 1 | wave 1 |
| 20-Mar-86 | S (3,5) | 71, 29, 0 | 1 | wave 1 | 13-Nov-86 | D (13,4) | 71, 29, 0 | 1 | wave 1 |
| 2-May-87 | U (0,0) | 33, 38, 19 | 1 | U | 2-Dec-87 | D (10, 8) | 38, 62, 0 | 1 | wave 1 |
| 6-Apr-88 | S (0,2) | 95, 5, 0 | 1 | wave 1 | 27-Oct-88 | D (13,2) | 100, 0, 0 | 1 | wave 1 |
| 15-Apr-89 | D (3,2) | 62, 38, 0 | 1 | wave 1 | 11-Nov-89 | S (2,5) | 90, 10, 0 | 1 | wave 1 |
| 8-May-90 | S (0,2) | 29, 67, 5 | 2 | wave 2 | 5-Dec-90 | S (4, 5) | 90, 10, 0 | 1 | wave 1 |
| 10-Apr-91 | U (0,0) | 86, 14, 0 | 1 | wave 1 | 15-Nov-91 | S (2, 10) | 14, 86, 0 | 2 | wave 2 |
| 22-Mar-92 | U (0,0) | 100, 0, 0 | 1 | wave 1 | 21-Nov-92 | S (0,1) | 57, 33, 10 | 1 | wave 1 |
| 12-Apr-93 | S (4,10) | 90, 5, 5 | 1 | wave 1 | 23-Nov-93 | D (6,4) | 29, 71, 0 | 2 | wave 2 |
| 2-Apr-94 | S (0,12) | 29, 67, 5 | 2 | wave 2 | 12-Nov-94 | D (17,3) | 100, 0, 0 | 1 | wave 1 |
| 8-Apr-95 | S (0,12) | 66, 14, 10 | 1 | wave 1 | 24-Nov-95 | U (8,8) | 76, 24, 0 | 1 | wave 1 |
| 10-Apr-96 | D (10,7) | 85, 10, 5 | 1 | wave 1 | 4-Dec-96 | D (11,8) | 100, 0, 0 | 1 | wave 1 |
| 30-Apr-97 | S (1,6) | 90, 10, 0 | 1 | wave 1 | 18-Nov-97 | S (9,12) | 57, 29, 14 | 1 | wave 1 |
| 29-Mar-98 | S (3,6) | 100, 0, 0 | 1 | wave 1 | 7-Dec-98 | D (6,3) | 48, 52, 0 | 2 | wave 2 |
| 3-May-99 | U (0,0) | 71, 29, 0 | 1 | wave 1 | 5-Dec-99 | D (12,0) | 67, 24, 10 | 1 | wave 1 |
| 10-Apr-00 | S (0,2) | 57, 29, 14 | 1 | wave 1 | 4-Nov-00 | D (20,5) | 100, 0, 0 | 1 | wave 1 |
| 10-May-01 | U (0,0) | 57, 14, 29 | 1 | wave 1 | 7-Dec-01 | S (1,8) | 24, 62, 14 | 2 | wave 2 |
| 4-May-02 | U (0,0) | 0, 86, 14 | 2 | wave 2 | 1-Nov-02 | D (19,4) | 76, 24, 0 | 1 | wave 1 |
| 14-Apr-03 | D (8,7) | 48, 52, 0 | 2 | wave 2 | 15-Nov-03 | S (0,2) | 62, 24, 14 | 1 | wave 1 |
| 30-Apr-04 | S (1,9) | 5, 86, 10 | 2 | wave 2 | 16-Nov-04 | S (6,8) | 29, 71, 0 | 2 | wave 2 |
| 13-Mar-05 | S (0,5) | 71, 29, 0 | 1 | wave 1 | 10-Nov-05 | D (12,7) | 90, 0, 10 | 1 | wave 1 |
| 7-May-06 | U (0,0) | 62, 0, 29 | 1 | wave 1 | 3-Dec-06 | S (6,15) | 43, 52, 5 | 2 | wave 2 |
| 19-Apr-07 | S (0,10) | 67, 33, 0 | 1 | wave 1 | 27-Nov-07 | D (6,1) | 76, 24, 0 | 1 | wave 1 |
| 1-May-08 | U (0,0) | 38, 43, 19 | 2 | wave 2 | 1-Dec-08 | D (5,0) | 71, 14, 10 | 1 | wave 1 |
| 10-May-09 | U (0,0) | 10, 76, 14 | 2 | wave 2 | 16-Nov-09 | D (11,6) | 52, 48, 0 | 1 | wave 1 |
| 30-Apr-10 | U (0,0) | 86, 14, 0 | 1 | wave 1 | 11-Dec-10 | D (5, 2) | 62, 29, 10 | 1 | wave 1 |
| 5-Apr-11 | S (3,4) | 86, 14, 0 | 1 | wave 1 | 25-Nov-11 | D (17,3) | 71, 29, 0 | 1 | wave 1 |
| 19-Apr-12 | S (5,6) | 29, 43, 29 | 2 | wave 2 | 6-Nov-12 | S (8,13) | 29, 71, 0 | 2 | wave 2 |
| 4-May-13 | U (0,0) | 67, 33, 0 | 1 | wave 1 | 2-Nov-13 | D (11,0) | 100, 0, 0 | 1 | wave 1 |
| 27-Mar-14 | S (0,2) | 38, 62, 0 | 2 | wave 2 | 22-Nov-14 | D (14,10) | 86, 14, 0 | 1 | wave 1 |
| 28-Mar-15 | S (0,3) | 86, 0, 14 | 1 | wave 1 | 11-Dec-15 | D (8, 3) | 71, 29, 0 | 1 | wave 1 |
| 5-Mar-16 | D (4,2) | 86, 14, 0 | 1 | wave 1 | 10-Nov-16 | D (12,10) | 38, 38, 24 | 1 | wave 1 |
| 8-Apr-17 | U (7,7) | 62, 14, 24 | 1 | wave 1 | 9-Nov-17 | D (21,10) | 100, 0, 0 | 1 | wave 1 |
| 15-Apr-18 | D (13,5) | 48, 33, 19 | 1 | wave 1 | 24-Nov-18 | S (3,10) | 90, 10, 0 | 1 | wave 1 |
| 23-Apr-19 | D (4,0) | 76, 24, 0 | 1 | wave 1 | | | | | |



*Author contributions.* The authors together initiated and designed the study. Both authors contributed to figures and both contributed to the
335  writing.

*Competing interests.* The authors declare that they do not have any competing interests.

*Acknowledgements.* A.B. was funded in part by NSF grant #1756958. Funding from the Swiss National Science Foundation to D.D. through
project PP00P2_170523 is gratefully acknowledged. We thank Dann Mitchell and William Seviour for providing code to calculate the vortex
elliptical moment diagnostics.




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
