# Peer review of "The Wave Geometry of Final Stratospheric Warming Events"

_Weather and Climate Dynamics, 2020_

## Community Comment (CC1)

**Comment**

January 16, 2021

**1  Comment on Authors' Definition of Final Stratospheric Warming in Southern Hemisphere**

**Author**: Nick Byrne

It's been well established in the literature (e.g. [1]) that ozone depletion has led to a delay (i.e. a positive trend) in the date of the final warming of the Southern Hemisphere stratospheric polar vortex, prior to the early 2000s at least. I was surprised by Figure 1c as there does not appear to be a visible trend in the authors' data. I performed some basic analysis to confirm this statement (see below), and also compared it against another definition that has been used in the literature ([1]). This analysis would appear to confirm that evidence for a positive trend is much weaker using the authors' data than in data previously used in the literature.

I wonder if this is because the authors use 10hPa in their definition for the final warming - perhaps this is a little high to use as a definition for the final warming? And given that the final warming in the Southern Hemisphere has been so closely tied to ozone depletion/a positive trend in the date of occurrence, I wonder if it might be confusing to introduce a new definition where this relationship is not so clearly visible?

On the other hand, I agree that it is attractive to be able to use the same level to define stratospheric events (FSW or SSW) in both hemispheres. I think there is a balance here between simplifying a definition and also maintaining the key properties attached to the original definition. Perhaps the authors could comment a little on where they think this balance lies? (There are convincing arguments for both sides. I think the most important thing is that the authors include some discussion in their paper about how their definition impacts previous work on long-term trends. Some related questions have also been raised by Reviewer 1.)

**1.1  Inspection of Data**

```
[1]:  # Day of year of FSW according to Butler & Domeisen definition.
      SFW_DATES_BUTL_DOM = [322, 322, 321, 323, 312, 310, 329, 317, 336, 300, 315,
                            339, 319, 325, 327, 316, 328, 338, 322, 341, 339, 308,
                            341, 305, 319, 320, 314, 337, 331, 335, 320, 345, 329,
                            310, 306, 326, 345, 314, 313, 328]
```

```
[2]:  # Day of year of FSW according to Black & McDaniel definition (see [1]).
      # NOTE: 2017 and 2018 datapoints have not been calculated
      SFW_DATES_BLA_MCD = [324, 326, 337, 333, 340, 335, 345, 339, 345, 322, 341,
                           348, 323, 341, 340, 328, 353, 344, 336, 356, 368, 331,
```

```
                    360, 337, 332, 332, 341, 350, 357, 358, 337, 355, 351,
                    324, 331, 347, 347, 323]
```

[3]: ```python
years = [1979 + x for x in range(40)]
```

[4]: ```python
from my_utils import plot_dates
```

[5]: ```python
plot_dates(years, SFW_DATES_BUTL_DOM, SFW_DATES_BLA_MCD)
```

[Figure]

[Figure]

**1.2 Trend Analysis**

We estimate the slope of the regression line, and its standard error, for subsequences of the full data. For example, the subsequence denoted by 1990 contains all FSW dates from 1979 - 1990 (inclusive). Similarly, the subsequence denoted by 2016 contains all dates from the original data, 1979 - 2016. (2017 and 2018 are not included as I do not have these datapoints for the Black & McDaniel time series.)

[6]: ```python
from scipy.stats import t, stats
```

Trend analysis for Butler & Domeisen timeseries:

```python
[7]: tinv = lambda p, df: abs(t.ppf(p/2, df))
```

```python
[8]: res_butl_dom = [stats.linregress(years[:i+2], SFW_DATES_BUTL_DOM[:i+2]) for i
                     in range(36)]
```

```python
[9]: slope_butl_dom = [x.slope for x in res_butl_dom]
     ci_butl_dom = [tinv(0.05, idx)*x.stderr for idx, x in enumerate(res_butl_dom)]
```

Trend analysis for Black & McDaniel timeseries:

```python
[10]: res_bla_mcd = [stats.linregress(years[:i+2], SFW_DATES_BLA_MCD[:i+2]) for i
                     in range(36)]
```

```python
[11]: slope_bla_mcd = [x.slope for x in res_bla_mcd]
      ci_bla_mcd = [tinv(0.05, idx)*x.stderr for idx, x in enumerate(res_bla_mcd)]
```

The blue dots represent the slope of the regression line (i.e. the trend) for a particular subsequence, and the black lines represent the 5% confidence thresholds. If a black line crosses the y=0 line, this means that the null hypothesis that the slope is zero (i.e. no trend) cannot be rejected at the 5% level for a given subsequence. This is a common test for checking for the existence of a trend in a timeseries.

```python
[12]: from my_utils import plot_trend_coef
```

```python
[13]: plot_trend_coef(years, slope_butl_dom, slope_bla_mcd, ci_butl_dom, ci_bla_mcd)
```

[Figure]

[Figure]

What the analysis indicates is that there is only one statistically significant subsequence using the definition of Butler & Domeisen whereas the more common definition of Black & McDaniel has many statistically significant subsequences i.e. evidence for an underlying trend is much stronger using the Black & McDaniel data.

**1.3 References**

[1] *Interannual Variability in the Southern Hemisphere Circulation Organized by Stratospheric Final Warming Events*, Black & McDaniel 2007, **JAS**

---

## Author Comment (AC1)

**Response to Reviewers**

We would like to thank the reviewers for their time and effort to review our manuscript. We here list the major changes that we made in the manuscript during the revision:

- To address the reviewers' concerns about the use of 10 hPa rather than 50 hPa dates, particulary for the SH, we now include wave classification for FSW dates at both levels in both hemispheres, using metrics at 50 hPa similar to previous studies (i.e., with a threshold of 5 m/s in the NH and 10 m/s in the SH). Results using these two levels are compared throughout the manuscript (or included in the Appendix). We think this provides a useful comparison of two common definitions for FSWs used in the literature.
- We no longer use moment diagnostics to calculate the classifications, since we agreed with the reviewers that these are not comparable to the wave forcing prior to the event. The classification is now solely based on metrics using wavenumber decomposition.
- In the process of updating this classification, we made some other changes that affected the revised wave classification. (A) Rather than using geopotential heights at different levels, the classification is performed using 50 hPa heights throughout (no matter the level where the FSW date was calculated). This is consistent with wave classification techniques for midwinter SSWs (our methodology now reproduces the classifications found for SSWs in Barriopedro and Calvo 2014, for example), and also accounts for the fact that wave-2 climatologically peaks at 50 hPa. (B) We decided to only use the 10 days prior to FSWs, rather than 10 days before and after. This isolates the wave forcing driving the event. (C) We use the full field geopotential heights to perform the wave decomposition, rather than anomalies. This better captures the physical wave structure of the vortex.

Detailed responses (in *cursive*) to the reviewer's comments and suggestions are below.

**Reviewer 1:**

This manuscript analyzes the wave geometry of Final Stratospheric Warming (FSWs) for both northern and Southern Hemispheres. It presents some interesting results that warrant publication. However, it is not clear to me that the differences between wave geometry are important (point 1 below). Also, the analysis of the Elliptical Diagnostics needs to be greatly improved or removed (major point 2). So, I think major revisions are required before the manuscript is suitable for publication.

We thank the reviewer for their comments, which we think have greatly improved the manuscript.

**MAJOR**

**1. Key results**

The presented analysis does find some differences between wave 1 and 2, but overall I find these differences to be small and even when there is a statistically significant difference it is not clear to me that these differences are important. Even the abstract doesn't present the results as having important implications. I actually came away with conclusion that probably not important /necessary to consider wave 1 vrs wave 2. This is an OK conclusion from a study, but I suspect authors disagree with this and if so they need to more clearly present results that will have implications.

I actually was more interested in some of the other results presented and only briefly discussed. In particular I think that there needs to be more discussion of trends, connection of SH FSW date with ozone, and late versus early. Even if some previously discussed you are presenting a longer time series. Also you could making comparisons with wave geometry.

The results have changed somewhat due to using new FSW dates and new classifications, but we have tried to emphasize more clearly in the abstract, results, and conclusions where the wave geometry is important. Additionally, we have added a new figure (Figure 2) linking the SH FSW to early spring ozone.

**For example:**

(a) The existence and lack of trends in Fig 1 is not discussed. Fig 1a appears to show a trend between 1979 and 2010 but I guess an insignificant trend over 1979-2019 or whole record. At least some comment on this is needed. Also possible trends / pause in SH needs to be mentioned (see more below).

We have addressed this by now including FSW dates at both 10 and 50 hPa, and comparing the trends (and correlations) between them. We have also added a new Figure 2, which shows the correspondence between FSW date and early spring (7 Sep- 13 Oct) total column ozone over the polar cap. We have also added text describing how the change in the trends in FSWs relates to ozone recovery and recent findings about changes in trends in the SH circulation (Banerjee et al. 2020).

(b) The connection between ozone and FSW needs to be expanded. The analysis here focuses on the impact of the FSW on ozone, but for SH (as stated in manuscript) a key point is timing of FSW is linked to ozone depletion. This connection is not actually shown here. Line 155 says SH final warming date tends to occur later during years of strong ozone depletion but no references are given ("As noted previously" is very vague). Even if noted before should be some analysis here. Also, on Line 232 it is stated that "These differences reflect a key point, which is that late events in the SH tend to occur in years with strong ozone depletion that further strengthen the vortex winds and allow the vortex to persist longer." Where is this shown in this paper? If this is a key point then show it. Returning to trends, does Fig 1b support a trend before 2000 and then a pause since (consistent with ozone recovery)? Banerjee et al reports such a pause in tropospheric circulation, is there also a pause in the FSW date trends?

We hope that our new Figure 2 and associated discussion addresses the reviewer's concerns about the connection between spring ozone and the FSW date. As far as the trends, the addition of the 50 hPa dates in the updated Figure 1 makes clear the trend towards later SH FSWs prior to 2000 and then a trend towards earlier SH FSWs since 2000. This is now discussed including the relationship to the Banerjee paper.

(c) I am not convinced that a wave separation provides any more (or even as much) insight as separation by timing. Fig 4 shows only small differences between wave 1 and 2, but on line 190 is states that "Early FSW events are associated with a stronger deceleration of the winds as compared to late FSW events due to the seasonally stronger winds earlier in the season". This needs to be shown and contrasted with the wave differences. Note Figure 5 also shows that the difference between late hrs early is more important than difference in wave geometry. Again, I don't think this means not valuable to show wave geometry differences but it just needs to be put in better context.

We have now added additional panels (e-h) to Figure 4 (which is now Figure 6), showing that the difference in wave geometry is smaller and less significant than the difference between early and late events when comparing the wind evolution in the stratosphere. We have also added an indication of when the winds for each type of event are significantly different from each other. Note that early and late events – by definition – contain a similar number of events, while wave-1 events strongly outnumber wave-2 events for both hemispheres, which adds to the fact that the difference between wave-1 and wave-2 events is less significant than the difference between early and late events. Additionally, the new Figure 5 which shows composite geopotential heights and anomalies for different classifications (both wave-1 and wave-2 as well as early vs late) hopefully provides some additional insight into how the wave geometry affects the structure of the stratospheric circulation, even if in the zonal-mean the zonal winds are not strongly affected. We also now make a point throughout the text to compare wave geometry to classification by timing, and to discuss where one seems more relevant than the other.

**2. Elliptical Diagnostics**

The fact that Elliptical Diagnostics (EDs) are considered is only mentioned in Section 2, and the fact they are used needs to be started earlier (and in conclusions). However, having said this, it is not clear to me that EDs are actually being used correctly or influencing the analysis. I think that additional analysis is needed or they should be removed.

Looking at table A2 it appears that for over half the NH cases and many of the SH cases the

ED classification does not agree with the final classification. In other words for many/most years the classification is due to agreement between the two wave number analyses but the ED are saying something else. This needs to be much more clearly presented (if EDs are kept).

The assumption of split = wave 2 and displacement = wave 1 is not valid, and you should not call a split a wave 2 or a displacement a wave 1. The fact this is not a good assumption is exactly the reason EDs were introduced. Further the disagreement between ED and wave numbers appears to support this.

If you keep the EDs then I think you should do an analysis based on S vrs D. Does this show any significance difference (is it similar / different to wave 1 vrs wave 2 difference).

The Elliptical Diagnostics are no longer used in the manuscript following the feedback from the reviewers.

**MINOR**

Line 102-105: Quote the standard deviation in date? Also, add a statement comparing April 12 and November 19 in terms of respective seasons (i.e. days after winter solstice / before summer solstice).

The standard deviations are 18 (15) days for the 10 (50) hPa classification in the NH (1958 - 2019) and 12 days at both levels for the SH (1979 - 2019). We have added this to the text. The median FSW in the NH at 10 (50) hPa occurs only 22 (25) days after the boreal spring equinox, but the median FSW in the SH at 10 (50) hPa occurs 57 (76) days after the austral spring equinox. We have also added this to the text, as well as a statement about the implications.

Line 109-110: Why not just use JRA for the complete record? It seems strange to add two reanlyses together and further complicates interpretation of the 1958-present day record.

We now use JRA-55 for the classification and analysis using the period from 1958 - 2019 for the NH and 1979 - 2019 for the SH given the uncertainties in the SH before the satellite era. We also list the ERA-interim classification in the tables where it deviates from the JRA classification.

Table 1 and 2: How about Table 1 = NH and table 2 = SH? I don't think any of the analysis compares the FSW in SH with that in NH for same year, and I think better to list all years for one hemisphere in a single table.

We have now divided the tables into NH (Table 1) and SH (Table 2) as recommended.

Line "133" "Another disadvantage" What was the first disadvantage?

As we no longer use this classification method, this sentence has been deleted.

**Short Comment (Nick Byrne):**

It's been well established in the literature (e.g. [1]) that ozone depletion has led to a delay (i.e. a positive trend) in the date of the final warming of the Southern Hemisphere stratospheric polar vortex, prior to the early 2000s at least. I was surprised by Figure 1c as there does not appear to be a visible trend in the authors' data. I performed some basic analysis to confirm this statement (see below), and also compared it against another definition that has been used in the literature (1). This analysis would appear to confirm that evidence for a positive trend is much weaker using the authors' data than in data previously used in the literature. I wonder if this is because the authors use 10 hPa in their definition for the final warming - perhaps this is a little high to use as a definition for the final warming? And given that the final warming in the Southern Hemisphere has been so closely tied to ozone depletion/a positive trend in the date of occurrence, I wonder if it might be confusing to introduce a new definition where this relationship is not so clearly visible? On the other hand, I agree that it is attractive to be able to use the same level to define stratospheric events (FSW or SSW) in both hemispheres. I think there is a balance here between simplifying a definition and also maintaining the key properties attached to the original definition. Perhaps the authors could comment a little on where they think this balance lies? (There are convincing arguments for both sides. I think the most important thing is that the authors include some discussion in their paper about how their definition impacts previous work on long-term trends. Some related questions have also been raised by Reviewer 1.)

We thank you for your comments, and for sharing your analysis which was helpful. We have now included 50 hPa dates as well as the 10 hPa dates. As you have shown, the SH 50 hPa dates capture much more strongly the trends associated with ozone depletion (and recovery). This is reflected in the updated Figure 1c and the new Figure 2. We also think the comparison of the dates at different levels is valuable, given that both are used in the literature. We have added discussion about why you might use dates at one level over another, and have also shown results for TCO anomalies and surface impacts using both definitions, to highlight where it makes sense to use the FSW defined at one level over another.

**Reviewer 2:**

This paper investigates whether final stratospheric warming events (FSWs) have a wave geometry associated with them, as found in midwinter sudden stratospheric warmings. They find that FSWs can be classified as wave-1 or wave-2 events fairly robustly using 3 different methodologies. They also find that such differences may be useful for improving tropospheric predictability, given that

each type can be associated with different near-surface signatures.

I found the paper to be interesting as it examines a topic that has mostly been overlooked in the literature and it is well-written. Although I would like to see more of a mechanistic explanation of some of the differences, particularly in the tropospheric response, which in the SH is quite startling (figure 6d,f), I feel that the paper is mostly good to go and worthy of publication. Hence, my suggestion is of minor corrections.

We thank the reviewer for their helpful comments. Note that because we changed how the dates/classifications for FSWs were calculated, our results for ozone and tropospheric impacts have also changed somewhat, but we do increase our discussion about how the wave geometry of the polar vortex could influence these differences.

Comments:

Line 31; Maybe add about the latest SH warming in 2019?

The analyzed events have been extended to include the 2019 FSW event in the SH.

Lines 109-110; It seems strange to stitch together the two reanalysis datasets, especially when JRA-55 could be used for both before and after 1979. Can you better explain why this is done? Why not just use JRA-55 for both periods (from 1979-2009, you say that dates vary only by 1-2 days, so surely it should not matter)?

We now use JRA-55 for the classification and analysis using the period from 1958 - 2019 for the NH and 1979 - 2019 for the SH given the uncertainties in the SH evolution before the satellite era.

Lines 100-110; I think something should be said about how your definition compares to other definitions. I understand that FSWs are usually defined using the 50hPa winds, which you mention is not possible in the SH as the winds do not clearly reverse from westerly to easterly at this level. But in the NH for instance, I wonder how the FSW classification at 10hPa and 50hPa may impact your results. I am not entirely sure, but I guess that the vortex breaks up at one level earlier than the other? This may affect your composites as you may be either capturing (or not) the actual wave pattern in the build up to the FSW. So in conclusion, I think that more should be said about the sensitivity to FSW definition; what if you define an FSW in NH using 50hPa winds, as I think is more traditional.

We have now revisited the classification of the FSW events in both hemispheres, as outlined in the overview to the Response. We now include comparison of FSWs defined at both 10 and 50 hPa (using thresholds previously used in literature). We also compare the composites, either in the main manuscript or in the Appendix, for both definitions.

Line 128; Is it the total number of days in [-10:10] spent with an aspect ratio or centroid latitude satisfying the threshold, or the longest continuous chain of days satisfied, that determines whether it is a wave-2 or wave-1? For instance, an event may overcome the aspect ratio threshold on day -10 but not again until days +7:+10, whereas the same event may overcome the centroid latitude threshold on days -1:2. I would say that the latter is the better classification (i.e., wave-1) as it occurs continuously around the actual FSW date. Further, what if the number of days are equal for the aspect ratio and centroid latitude fulfilment?

As we no longer use this classification method, this sentence/description has been deleted. However, for the wave decomposition method, we did change the classification to only consider the 10 days prior to the event (rather than the 10 days prior and after) to focus on the wave forcing right before the event.

Lines 145-151; Just to clarify, when you say 50-hPa classification here, you only mean for determining wave-1 and wave-2, right? (This is valid for the whole manuscript as it can sometimes be a bit confusing). If not, then my point just above will be redundant!

We have now clarified this in the manuscript with the revised event classification. The wave classification is now always performed using 50 hPa geopotential heights. This is more consistent with previous classifications using wave decomposition for midwinter SSWs (Barriopedro and Calvo 2014).

Figure 1; I won't go into details here as I realise that both Darryn and Nick have mentioned this substantially in their reviews/comments, but I was aware that there is a well-known trend in the SH final warming dates, linked to ozone depletion. This trend is lacking in fig. 1c and I am wondering why. As Nick states, it may be related to the FSW definition, so discussing the sensitivity of your dates to the level of FSW definition, as aforementioned in a previous comment of mine, would be useful.

We have now included 50 hPa dates as well as the 10 hPa dates, which allows us to compare the sensitivity of our results using the two different definitions. As suggested by the other reviewers, the SH 50 hPa dates capture much more strongly the trends associated with ozone depletion (and recovery). This is reflected in the updated Figure 1c and the new Figure 2.

Figure 2; What do the composites look like? Do they show typical wave-1 and wave-2 patterns, or does the longitudinal location of the vortices vary too much between each FSW so that there is

cancellation? During SSWs, the vortex locations are pretty constant between events, so I wonder here if it is similar during FSWs. I also think it would be good to show this to see if there are indeed statistically-significant differences between the wave-1 and wave-2 FSWs.

Thanks for this suggestion. We have added a figure showing the composites of 50 hPa geopotential heights and anomalies for different dates and classifications (new Figure 5). We think this very clearly illustrates where the vortex geometry is consistent for different events and where it is more variable.

Line 170; 'positive'  $\rightarrow$  do you mean ratios greater than one?

This sentence has been deleted in the revision.

Line 181; Not strictly a deceleration as has units of m/s. It is just the difference between the two time periods right?

Thanks for spotting this, we have corrected this in the manuscript.

Lines 183-190; Are the differences between the wave-1 and wave-2 events statistically significant?

Significance levels have been added to all panels in Figure 4 (now Fig 6).

Lines 193-194; I cannot see this from the plot. It appears that the mean lines (thick) cross zero at 50hPa after 20days or so (panel c). Is this what you are referring to?

This sentence has been deleted in the description of the updated figure.

Figure 5; Can you perhaps include a PV contour marking the 'edge' of the polar vortex to see how it compares with the ozone anomalies?

The PV gradient/vortex edge becomes wavy and washed out right around the spring transition (especially for late events); this was one issue with using the moments diagnostics. However, we think the new composite map of geopotential height contours (new Figure 5) in comparison to the updated ozone anomaly plots (now Figure 7) demonstrates strong correspondence, which is described in the text.

Lines 231-232; Can this be shown explicitly? Figure 5 does suggest it, but I would rather it be shown. For instance, you could calculate the years that have stronger ozone depletion compared to some mean and compare and perhaps calculate a correlation with the years that have a late

FSW. Just a suggestion for a figure, but I think this point could be better shown in the manuscript.

We have added a new Figure 2, which shows the correspondence between FSW date and early spring (7 Sep- 13 Oct) total column ozone over the polar cap. This clearly demonstrates a correlation between years with stronger ozone depletion and later FSWs.

Lines 260-264; It appears to me that the wave-1 and wave-2 events lead to nearly identically opposite 500-hPa GPH anomalies in the SH. Any idea why this is? Is this also the case at levels above, perhaps in the stratosphere? Figure 5g,h did not show such an opposite-signed pattern in the TCO. Further, are there significant differences lower down, perhaps in the MSLP?

With the updated analysis (changes in classifications/dates), the patterns are no longer opposite (now Fig 8) except when using dates at 10 hPa (see Fig A2), and the anomalies associated with wave-1/wave-2 are generally insignificant using dates at either level, hence this might have been an artefact. Differences in surface impacts are much more apparent for early vs late SH FSWs, which we now include. Wave-1 and wave-2 are associated with significant TCO anomalies in the SH (now Figure 7), but keep in mind these anomalies are for the 10 days prior to the FWs, whereas the tropospheric impacts are for the 7-30 days after.

---

## Author Response (AR2)

**Response to Reviewers**

We would like to thank the reviewers for their time and effort to review our manuscript. Detailed responses (in *cursive*) to the reviewer's comments and suggestions are below.

**Reviewer 1:**

The revisions satisfy my original concerns and I think have greatly improved the paper, and it is acceptable for publication as it is.

However, I think the authors should consider revising the last sentence of the abstract. This could be read to say the the timing of the FSW determines the level of ozone depletion (as "anomalies" is vague), whereas it is the other way around.

*Comments: We are glad that we have satisfied the reviewer's concerns. We have re-written the last line of the abstract as follows. "In the Southern Hemisphere, the timing of the FSW is strongly linked to both total column ozone before the event and the tropospheric circulation after the event."*

**Reviewer 2:**

I am happy with the changes made to the manuscript in response to my comments and those of the previous reviewer/commenter. The paper is interesting and I think can be useful for dynamicists and forecasters. Aside from a couple of very minor points below, my suggestion is to publish as is.

*Comments: We thank the reviewer for their comments and we are glad they are happy with the revision. We address the remaining minor comments below.*

Figure 2: It may make more sense to include this in section 4 where the independent ozone dataset is introduced.

*We appreciate the suggestion and considered moving this figure, but it seems important to be able to explain the differences in the SH 50 hPa and 10 hPa dates, and the differences in NH and SH processes, in Section 2. We do cite the dataset in the Figure 2 caption, and have added "see Section 4" to the caption as well.*

Figure 5; I think performing a 2-sided t-test on the difference between the individual pairs of panels would be more useful given that you compare in the text, early vs late FSWs and wave-1 vs wave-2

dates (e.g., lines 190-201, 207-211). Currently you only show stippling for significantly different values from zero, but actually it is the comparison between panels that is important. I guess the 10hPa vs 50hPa differences are less important, but the wave-1 vs wave-2 differences are important for the main objective of the paper (to classify and document differences between such FSW types).

*We appreciate this comment. One of the goals of Figure 5 in particular was to determine where the response was robust across events for different types of classifications (i.e., do wave-2 events always or often occur in the same way spatially?). We think Figure 5 gives an indication of the commonality across events of certain features of a wave-1/2 event in particular. Therefore we think for this figure it makes sense to use a statistical test to test the significance from zero. However, we agree that later on we are particularly interested in comparing the differences in the impacts of FSW geometry and timing. So, for Figures 7 and 8 (and corresponding Appendix figures) we have changed the stippling (except in panels a,f which show the composite for all events) to the significance between, e.g., wave-1 and wave-2 or early vs late. We agree that this helps the reader visualize where the differences may be most important. Small changes to the text in reference to statistically significant differences have been added.*

Line 271-272; Although it is worth mentioning that this is not robust and currently up for debate. I have seen two papers in the last month on this issue, both in JGR atmospheres; Hall et al. (2021) and White et al. (2021). The former examined observations and the latter used an idealized model. Both found that differences between displacements and splits (wave-1 and wave-2 events) were only present at lags close to the onset. At longer lags, differences are not apparent.

*We thank the reviewer for pointing this out. We do bring up some of these issues in the conclusions, but we have also changed this line to the following and included the suggested references: "There are observed differences in the NH surface impacts following displacement and split-type SSW events (Mitchell et al. 2013), though the robustness of these impacts are debated (Maycock and Hitchcock 2015, White et al. 2021, Hall et al. 2021)."*